

# Disturbed flow in an aquatic environment may create a sensory refuge for aggregated prey

Asa Johannesen[1,2], Alison M. Dunn[2] and Lesley J. Morrell[3]

[1] Nesvik Marine Centre, Fiskaaling, Hvalvik, Faroe Islands
[2] School of Biology, University of Leeds, Leeds, United Kingdom
[3] School of Environmental Sciences, University of Hull, Hull, United Kingdom

## ABSTRACT

Predators use olfactory cues moved within water and air to locate prey. Because prey aggregations may produce more cue and be easier to detect, predation could limit aggregation size. However, disturbance in the flow may diminish the reliability of odour as a prey cue, impeding predator foraging success and efficiency. We explore how different cue concentrations (as a proxy for prey group size) affect risk to prey by fish predators in disturbed (more turbulent or mixed) and non-disturbed (less mixed) flowing water. We find that increasing odour cue concentration increases predation risk and disturbing the flow reduces predation risk. At high cue concentration fish were able to locate the cue source in both disturbed and non-disturbed flow, but at medium concentrations, predators only located the cue source more often than expected by chance in non-disturbed flow. This suggests that objects disturbing flow provide a sensory refuge allowing prey to form larger groups, but that group sizes may be limited by level of disturbance to the flow.

## INTRODUCTION

To avoid predation, animals use strategies from visual crypsis (*Jackson et al., 2005*) to increased vigilance in groups (*Krause & Ruxton, 2002*). In cases where visual interactions between predators and prey are limited, cues such as sound (*Obrist et al., 1993*) or detection of electric fields (*Kajiura & Holland, 2002*) are used instead. Olfaction is a key sense used in prey detection and location. Olfactory predators such as crustaceans (*Gomez & Atema, 1996*; *Weissburg & Zimmer-Faust, 1993*), fish (*Nevitt, 1991*), and molluscs (*Ferner & Weissburg, 2005*) can successfully track odour plumes from prey to their source. Animals use many sensing strategies, including time differences in bilateral odour detection (*Gardiner & Atema, 2010*), time-averaging of odour concentrations (*Wilson & Weissburg, 2012*), and making simultaneous comparisons of odour concentration (*Page et al., 2011*; *Vergassola, Villermaux & Shraiman, 2007*). To avoid such detection, prey may try to limit the amount of olfactory cue that they release or otherwise make it difficult for predators to detect them (*Ruxton, 2009*).

Corresponding author
Asa Johannesen, asajoh@fiskaaling.fo

To reduce the risk from predators that hunt using vision, prey can group together to increase the time taken for a hunting predator to locate them. This is known as the encounter-dilution effect (*Wrona & Dixon, 1991*) and favours grouping as an anti-predator strategy in response to visual predators (*Jackson et al., 2005*; *Riipi et al., 2001*). However, if increasing group size makes prey increasingly easier for olfactory predators to find (*Kunin, 1999*), grouping may be counter-productive.

Larger or more numerous animals release more odour cue, eliciting a stronger reaction in the receiver (*Hawkins, Magurran & Armstrong, 2007*; *Kusch, Mirza & Chivers, 2004*). When animals aggregate, the odour cues released interact, increasing the size and concentration of the odour plume (*Villermaux & Duplat, 2003*), which allows receivers to more readily track the plume to its source (*Wilson & Weissburg, 2012*). Grouping benefits prey avoiding olfactory predators in still water (*Johannesen, Dunn & Morrell, 2014*). However, water movement provides a directional cue to the prey, so olfactory cues are more easily taken advantage of in flowing rather than still water (*Løkkeborg, 1998*).

In a review of olfactory detection distance in insects, *Andersson, Löfstedt & Hambäck (2013)* indicate that the increase in detection with increasing size of the source is likely to be asymptotic, although theoretical work indicates accelerating detectability may also be possible (*Treisman, 1975*). If the risk of predation increases too much with group size, aggregation would be counterproductive in species that cannot otherwise defend themselves. Here, we explore this question from the perspective of three-spine sticklebacks (*Gasterosteus aculeatus*) locating odour sources of differing concentration (as a proxy for prey group size (*Hill & Weissburg, 2014*; *Schneider et al., 2014*)—but see discussion) in flowing water, to test the hypothesis that increasing prey (bloodworm) cue concentration increases the risk to prey in flowing water.

Chemical cues are often detected in pulses because currents, turbulence, and other types of disturbed flow create patches of cue (*Finelli & Pentcheff, 1999*; *Weissburg & Zimmer-Faust, 1993*; *Zimmer-Faust et al., 1995*), which may create 'sensory refuges' (*Weissburg & Zimmer-Faust, 1993*). When prey are in these refuges, predators may be less- or unable to detect them (*Ferner & Weissburg, 2005*), while prey may still be able to detect predators as back eddies carry odour cues 'upstream' (*Dahl, Anders Nilsson & Pettersson, 1998*). Prey animals occupying a sensory refuge would benefit from the reduced predation success, leading to aggregation of prey in refuge areas. If animals aggregate in sensory refuges, the sensory refuge may counteract the increased risk of detection due to larger group size. We repeat our experiment in disturbed flow to examine the additional hypothesis that disturbed flow reduces the risk to prey relative to undisturbed flow, as it creates sensory refuges (*Weissburg & Zimmer-Faust, 1993*). Our aim is to provide an initial exploration of the possible impact of prey aggregation and flow conditions on the detection of prey by a foraging fish.

## METHODS

### Experimental species, transportation and housing

Two hundred three-spine sticklebacks *Gasterosteus aculeatus* (4–5 cm total body length) were caught in a pond in Saltfleet, Lincolnshire in November, 2011 (53°25′59.55″N,

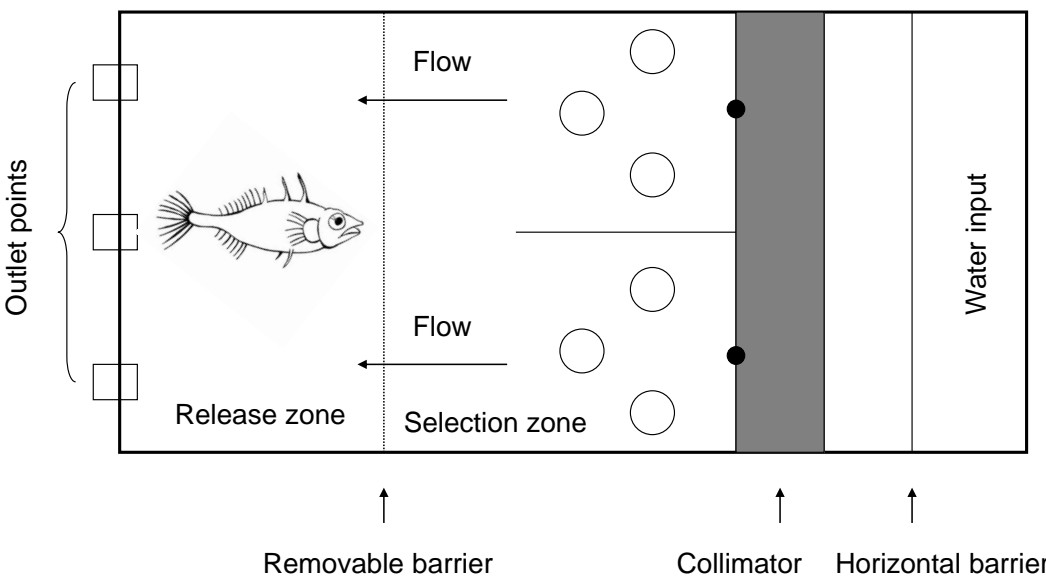

**Figure 1 Layout of the Y-maze (Total dimensions: length 93 cm and width 40 cm).** Water flowed over the horizontal barrier (mid-way in the 30 cm long header chamber) before entering the Y-maze through a collimator (10 cm long) to ensure even flow on both sides. The water flowed along the arms of the Y (20 cm long) before entering the stem (33 cm long), which was partitioned with a removable barrier for the release zone (20 cm long). Water flowed through the Y maze at approximately 3 cm s$^{-1}$ before exiting through the outlet holes (3 cm in diameter). Cue input points are marked by a black dot. Large open circles represent the cylinders (5.5 cm in diameter) added to the tank in the 'disturbed flow' treatments.

0°10′49.41″E) and transported in fish bags (3–5 fish per litre) to aquarium facilities in Leeds (3 h journey). Sticklebacks can detect and locate prey using non-visual cues in still water (*Johannesen, Dunn & Morrell, 2012*), but also occupy flowing water environments and may be able to track odour plumes to their source. Fish were housed in grey fibreglass tanks (0.5 × 0.5 × 1.0 m) with gravel substrate, plastic plants, plant pots and mechanical filters. Light regime was 10:14 h light:dark, temperature was 14 ± 2°C and fish were fed daily on defrosted frozen bloodworm. Fish were kept for six months to one year for experimentation prior to release where caught in agreement with the Home Office and Defra.

## Procedure

Trials were carried out in a flow-through rectangular tank (40 cm by 53 cm, water depth 9 cm, flow velocity 3 cm s$^{-1}$) based on a Y maze design (*Ward, Herbert-Read & Simpson, 2011*) (Fig. 1). The stem of the maze was 33 cm in length including a 20 cm 'release zone' with a removable barrier. Each 'arm' was 20 cm long. Conditioned water was pumped from a header tank into the maze, entered the maze over a horizontal barrier in both arms of the Y, and passed through a collimator to create an even water flow ("undisturbed flow"). Flow characteristics were not measured, but pilot trials using dye indicated that the odour cue would move through the water evenly. Water left the flume through 3 mesh-covered exit holes evenly spaced across the base of the stem of the Y, and was not re-circulated. Trials were observed from behind a screen via a webcam to reduce disturbance to the fish.

Olfactory cues were created using filtered macerated bloodworm (cue concentrations; low: 5 g l$^{-1}$ medium: 10 g l$^{-1}$ and high: 20 g l$^{-1}$). Although spatial distribution of aggregated prey affects odour plume composition, we use prey cue concentration as a proxy for size or number of prey. This is because the interacting odour plumes from multiple prey individuals will increase cue concentration (*Hawkins, Magurran & Armstrong, 2007*; *Villermaux & Duplat, 2003*) and while adding the spatial factor to this experiment would be more realistic, it would also complicate the experiment. Cues were delivered to the maze using two peristaltic pumps at a rate of 10 ml min$^{-1}$ (source diameter 4 mm, velocity 1.3 cm s$^{-1}$). In each trial, the olfactory cue entered at one arm of the maze, and a conditioned water control (containing red food dye to copy the tint of the bloodworm cue water) entered at the other at the same rate. Cue side was allocated at random to control for side preference. After the trial, the maze was emptied and refilled with conditioned water to remove olfactory cues from the previous trial.

Sticklebacks were placed individually into the release zone and allowed to acclimatise until they resumed normal behaviour (start—stop swimming at moderate speed, five minutes minimum). Following acclimatisation, the water inlet pump was switched on and ran for two minutes (to stabilise flow) before cue pumps were turned on. The behaviour of the test fish was monitored until it had visited both sides of the stem of the Y (two minutes minimum) and the barrier was raised using a pulley system. The fish was allowed five minutes to reach the top of one arm of the Y, where its choice (cue or control) was recorded. The time taken for the fish to acclimatise (begin swimming) and the time taken to reach the top of the chosen arm were also recorded. Fish were excluded from the experiment if they did not resume normal behaviour in the release zone ($N = 23$), did not visit both sides of the stem of the Y within 5 min ($N = 8$ fish) or did not make a choice ($N = 6$). Final sample sizes in undisturbed flow were: low: $N = 16$, medium: $N = 16$, high: $N = 16$.

We subsequently investigated the effect of disturbed flow on stickleback choice in the maze. Three cylinders were placed in each arm of the Y maze to create downstream disruption to the flow (see Fig. 1). Visualisation of the flow using food dye indicated that the cylinders caused the odour plumes to split and disperse, and that the plumes appeared qualitatively different to those in the experiment with no disturbance to the flow. Methods were the same as in the previous experiment, but investigated only two cue concentrations: medium and high. The low concentration was not used as the first experiment indicated that fish did not show a preference at this concentration (see results). Eight fish were excluded from this experiment, giving final sample sizes of $N = 17$ for medium cue concentration and $N = 17$ for high cue concentration in disturbed flow. Each fish was used only once in the experiments, and different fish were used in the two flow conditions to avoid any learning effects. The total sample size for both experiments was $N = 82$.

## Analysis

Data were analysed using R v 2.13.0 (*R Core Team, 2013*). Cox proportional hazards survival models were used to analyse fish time to acclimatise and time to choose (survival package in R; (*Therneau & Lumley, 2011*)). Survival models are highly flexible and useful

for time-to type data, especially when data do not follow a Gaussian distribution and contain censored times (*Therneau & Grambsch, 2000*). Preference for prey cue was tested using binomial exact tests against an expectation of random (0.5). A correction for false discovery rate was performed (*Benjamini & Hochberg, 1995*) on the tests to control for multiple comparisons and the adjusted p values are given in the results section. A binomial GLM with choice as dependent variable (cue, no cue) and cue concentration and flow type as independent variables was used to test the effect of treatments on choice.

### Ethical statement

As experiments with fish fall outside of the remit of the University of Leeds Ethical Board and no licensed procedures were used, this study was not subject to ethical review.

However, laboratory experiments were carried out in accordance with University of Leeds guidelines and in agreement with Home Office licensed technical staff at the animal facility. Great care was taken to ensure optimal welfare for all fish involved in this study.

## RESULTS

Fish tested in the disturbed flow condition took less time to acclimatise than those in the non-disturbed flow condition (coxph: Chi-squared $= 25.81$, $df = 1$, $p < 0.001$), but there was no effect of cue concentration or flow type on time to choose once acclimatised (coxph: Chi-squared $= 6.22$, $df = 5$, $p = 0.29$).

In the 'undisturbed flow' condition, fish selected the cue arm over the control arm at medium ($N = 13/16$, P(success) $= 0.8125$, $p = 0.035$) and high ($N = 15/16$, P(success) $= 0.938$, $p = 0.003$) cue concentrations, but not at the low cue concentration ($N = 11/16$, P(success) $= 0.688$, $p = 0.26$). When flow disturbance was added, fish preferentially selected the cue arm at high ($N = 14/17$, P(success) $= 0.824$, $p = 0.033$) but not medium ($N = 10/17$, P(success) $= 0.588$, $p = 0.629$) cue concentrations (Fig. 2).

In a test of the effects of cue concentration and flow regime, cue concentration significantly affected choice (Binomial GLM; $z = 2.235$, $N = 82$, $p = 0.025$) while the effect of water regime fell short of significance (Binomial GLM; $z = -1.814$, $N = 82$, $p = 0.070$).

## DISCUSSION

Our results suggest that in a Y-maze with olfactory cue presented in one arm only, fish predators can successfully choose the arm containing the cue more often than expected by chance if the concentration of the cue is high enough. At the low cue concentration, fish did not choose the cue arm more often than the non-cue arm. Interpreted in the context of our question of how increasing group size affects detection, this suggests that grouping in prey (increased cue concentration) may increase risk from olfactory predators. Adding objects to the maze to disturb the flow (i.e., create more turbulent mixing) decreased the number of successful choices, particularly at the medium cue concentration, suggesting that 'sensory refuges' created by disturbed flow (*Weissburg & Zimmer-Faust, 1993*) allow larger groups to form by countering the increased risk of detection. However, at higher cue
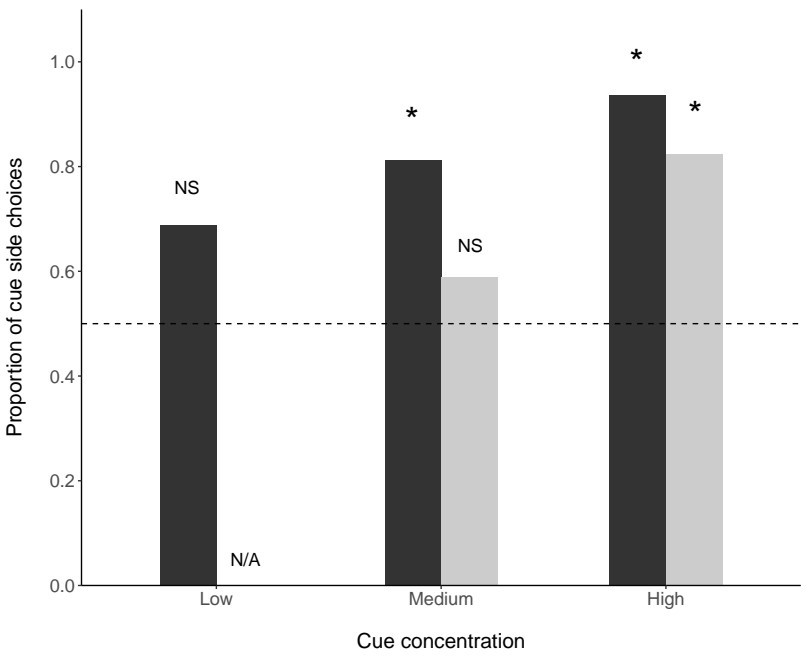

**Figure 2  Proportion of fish choosing the prey side in a y-maze.** Stars above bars signify significant differences (binomial exact tests) from random choice of side. Dark bars are non-disturbed flow treatments and light bars are disturbed flow treatments. The horizontal dashed line indicates no preference.

concentrations, the benefit gained from flow disturbance may decrease, and larger groups would be more easily detected. While we did not compare multiple disturbance levels, we suggest that the level of disturbance may also influence the detection of prey cues of different concentrations.

We used concentration as a proxy for group size, following previous work (*Schneider et al., 2014*), although the effect of group size on odour plumes is more complex. When individuals group together, they produce a greater number of odour filaments (*Monismith et al., 1990*; *Wilson & Weissburg, 2012*) that cover a wider area (*Webster & Weissburg, 2001*). When prey are spaced sufficiently far apart, individual odour filaments do not interact and would not result in an increase in cue concentration. In that case, the risk of detection by predators who are unable to detect any individual filament, but would detect a more concentrated one, remains low regardless of group size. However, in cases of tightly aggregated prey, filaments are likely to interact and increase the time-averaged concentration (*Villermaux & Duplat, 2003*) used by some predators to track plumes (*Ferner & Weissburg, 2005*) as well as filament concentration (*Villermaux & Duplat, 2003*). Therefore, tightly packed prey increase risk of detection with group size, particularly by predators who would not have detected a single individual's odour plume.

Concentration may reflect a number of other features of the prey landscape in addition to group size in tight aggregations. As cues become more diluted over distance and time, cue concentration may signal distance to prey (but see (*Bytheway, Carthey & Banks, 2013*)), or if larger individuals release more cue, the concentration could signal size of prey. Thus, predators could move towards higher concentrations because they represent better value

predation opportunities, rather than because they are easier to detect. However, while the results in this experiment could reflect a perceived value of cue in the fish rather than simple detection, there is no indication of this in the latency to choose a cue arm in our data.

While turbulence or other disturbance to flow can cause odour plumes to break up (*Webster & Weissburg, 2001*) it can also act to mix the plumes and dilute the cue to background levels with only intermittent spikes (*Webster & Weissburg, 2009*) that may not be worth exploring. Either mechanism would act to make tracking the cue to the source more difficult for the predator (*Robinson, Smee & Trussell, 2011*), although this may depend on the predator's sensing strategy and sensitivity (*Ferner & Weissburg, 2005*). Our observations with food dye suggest that the plume in disturbed flow split primarily into two meandering plumes. Assuming a fish was only exposed to one arm of the split plume, the decreased amount of cue could mean fewer or smaller prey, greater distance to prey, or the concentration might be below a detection threshold. A meandering plume will, in addition to the perceived lower reward, be more difficult to track for fish and other filament samplers, making the effort greater. While a time-averaging predator may be able to compensate for meandering plumes (*Page et al., 2011*), a diluted and split plume signals lower reward, so inhabiting turbulent water can also benefit tightly packed prey hiding from a time-averager. However, as sticklebacks most likely do not use time-averaging sensing strategies (*Nevitt, 1991*; *Webster & Weissburg, 2001*), this has not been explicitly tested here.

Our study did not investigate the fluid mechanics and transport of olfactory cue in the different flow regimes, focusing instead on the response of the predator. Thus, we cannot speculate on the sensory mechanism, motivation, or features of water flow and cue transport that cause the different behaviours shown by our fish predators. However, the end result for the prey remains the same. If an olfactory prey cue is highly concentrated, indicating either great reward (many or large prey) or easy reward (close proximity) a predator is more likely to pursue that cue. Conversely, if the olfactory prey cue plume is somehow broken down, indicating small reward (few or small prey) or difficult reward (long distance, a plume that is difficult to track) a fish predator is less likely to pursue that cue. In the context of our question regarding aggregation, this suggests prey are able to aggregate into larger groups by taking advantage of a sensory refuge and either fooling the predator into thinking they are not worth the effort (small reward/high effort) or decreasing the cue to avoid detection. Individuals in such aggregations would in turn benefit from greater survival chance if found due to other benefits of grouping, such as predator satiation.

The study of anti-predator aggregation has primarily focused on predators that use vision to detect their prey (*Ioannou & Krause, 2008*), while the effect of olfactory predators on the evolution of aggregation is less well understood. Our work suggests that group size may interact with environmental parameters, and that the evolution of grouping in response to olfactory predators may be dependent on the flow environment. However, further work is needed to fully investigate the relationship between grouping prey, detection by predators, and environmental conditions. Prey are known to aggregate in streams (*Rasmussen &*

*Downing, 1988*), but aggregation decisions may depend on factors other than risk from olfactory predators, including foraging opportunities, flow speed and risk from predators relying on other sensory modalities (*Ioannou & Krause, 2008*; *Krause & Ruxton, 2002*). Experimental manipulation and characterisation of flow regimes and the response of predators and prey may help disentangle the interacting effects of group size, flow regime and aggregation in response to other resources.

### Funding

This work was supported by the Faroese Research Council. The funders had no role in study design, data collection and analysis, decision to publish, or preparation of the manuscript.

### Grant Disclosures

The following grant information was disclosed by the authors:
Faroese Research Council.

### Competing Interests

The authors declare there are no competing interests.

### Author Contributions

- Asa Johannesen conceived and designed the experiments, performed the experiments, analyzed the data, contributed reagents/materials/analysis tools, wrote the paper, prepared figures and/or tables.
- Alison M. Dunn reviewed drafts of the paper, discussed methodology and made useful suggestions for experimental design.
- Lesley J. Morrell conceived and designed the experiments, reviewed drafts of the paper.

### Animal Ethics

The following information was supplied relating to ethical approvals (i.e., approving body and any reference numbers):

As experiments with fish fall outside of the remit of the University of Leeds Ethical Board and no licensed procedures were used, this study was not subject to ethical review.

However, laboratory experiments were carried out in accordance with University of Leeds guidelines and in agreement with Home Office licensed technical staff at the animal facility. Great care was taken to ensure optimal welfare for all fish involved in this study.

### Data Availability

Johannesen, Asa (2016): Y-maze data for flow experiment. figshare.
https://doi.org/10.6084/m9.figshare.985515.v1.

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
