# Peer review of "Disturbed flow in an aquatic environment may create a sensory refuge for aggregated prey"

_PeerJ, doi:10.7717/peerj.3121_

## Round 0.1 · original submission · Major Revisions

· Academic Editor

Major Revisions

Reviewer 1 judged the MS to need a Major revision, Reviewer 2 judged it to need a Minor revision. I fall in between, but feel confident that you can successfully revise the MS to be acceptable. Both reviewers made comments to help with a revision. Reviewer 1 provided a pdf with comments and suggestions. I downloaded Reviewer 1's pdf and made my comments on there as well (mine are marked with my initials MEH). In general you need to:

1) Improve your writing style to increase the precision and economy of word usage.
2) Deal with ambiguities of aggregation (which has a spatial component) and and concentration (which in your experimental set-up does not). I made notes on the pdf with suggestions for this.
3) carefully consider Reviewer 1's concerns and suggestions about statistical procedures.

Reviewer 1 ·

Basic reporting

The work is generally well situated and adequately discussed. However, because the article is framed (in my opinion, improperly) as an investigation of the potential role of aggregation, they have missed some basic literature. This will need correction.

The authors have a tendency to use clauses excessively, creating somewhat cumbersome sentences. I have made a few initial suggestions, but the authors will want to go through the ms carefully to simplify things as needed.

Experimental design

The research question is more limited than claimed-it is really about the role of turbulent processes in modulating olfactory search. This process has not been extensively examined in fish although this has been studied in other organisms. The study makes a contribution here, but the basic finding has been documented numerous times in other groups.

The methodology is not consistent with the framing of the question. The authors argue concentration is a proxy for aggregation, which is only partially true. Aggregation results in differing patterns of inter-individual spacing and overall size that will affect the signal and response. Indeed, one of the cited papers (Wilson) makes this point. Such factors are not accounted for in the experimental design, which manipulates only flow regime and concentration. This reviewer therefore thinks the weak relationship does not merit framing this study as an investigation of the role of aggregation. It is really about the effects of turbulence on fish responses to chemical plumes and should be motivated and discussed from this perspective. The connection to aggregation can be part of this, but it should not be central. The authors will need to include more literature on basic processes animals use to determine responses to chemical plumes and the role of turbulence.

Validity of the findings

The statistical methods are not appropriate as far as I can tell. The authors perform a series of single df comparisons on data that come from a single experiment. It would be best if the authors would use a log-liklihood model to examine the effects of concentration, flow and their interaction, as well as to pinpoint individual differences. One might also use a parametric approach if the explicit comparison to the no-choice (e.g. 50:50) outcome is not required. The authors will need to apply some sort of post-hoc correction (such as Bonferroni) if they insist on individual comparisons. It may be that it is better to use different analysis to test the deviation from the null vs. the role of concentration and flow regime.

As argued above, much of the current discussion of how the results relates to aggregation need to be reduced and tempered by the observation that the experimental design is not an explicit test of the role of aggregation and fluid regime.

Additional comments

I've made some explicit suggestions and comments on the ms that are relevant to the points in the review.

Annotated reviews are not available for download in order to protect the identity of reviewers who chose to remain anonymous.

Reviewer 2 ·

Basic reporting

I found the article to be well written and interesting. It only needs some very minor reworking before publication.

Experimental design

The experiment is correctly designed, has appropriate replication, and the analysis is appropriate.

Validity of the findings

The data is robust and statically sound. The conclusions are well supported.

Additional comments

The authors have performed an interesting study to determine if flow variation can mask chemical signals from prey, allowing prey to form larger aggregations. I found the paper very interesting and recommend it for publication. Overall the paper is well written. Below I offer a few ideas to consider for improvement. These issues are minor and the article should be accepted with these slight changes.

Authors use English spelling throughout (e.g., stabalise instead of stabilize, odour instead of odor). Not sure what PeerJ’s policy is on that. Also, in many instances, the authors have two sets of parentheses. Usually I see one set with a comma. For example, Line 39 (e.g., sharks (Gardiner and Atema, 2010)) should probably be (e.g., sharks, Gardiner and Atema 2010).


Line 17-18 please reword. How is release of odor cues related to group size? I think what is meant here is that the amount of prey cue released increases with increasing prey numbers, likely making the group more recognizable to predators.
Line 19 Please add the word ‘may’ after flow, in some cases, turbulence may increase foraging success
Line 21 and throughout. The flows on this scale are not really laminar. I recommend rewording to focus more on the flow disruption. Perhaps restating laminar as undisturbed flow as you do on line 24? Please do this not only here but throughout. Laminar flows really cannot occur at these Reynolds numbers.
Line 24-26 delete ‘downstream of prey’ I don’t think it matters where the disruption occurs. I suggest simply rewording to say that disturbed flows may mask prey chemical cues and create a sensory refuge promoting aggregation. Also, the abstract is missing the interaction between cue concentration and flow. In the highest cue concentration, predators still selected followed the prey cue in disturbed flow. The relationship between aggregation size, cue concentration, and degree of disturbance would be useful here.
Line 164 laminar flow, please reword
Line 188 add the after low to read in At the low cue concentration…
Double period on line 195

The discussion would also benefit from a few lines discussing the relationship between disruptions in flow and the level of aggregation possible. In other words, there seems to me to be a trade-off between level of aggregation possible and magnitude of flow disruption.

End of Review

---

## Round 0.2 · Minor Revisions

· Academic Editor

Minor Revisions

Your MS was returned to only the more negative reviewer for another viewing. That reviewer found the MS improved, but suggested some slight revisions. Please look over the review, make changes where they are needed or improve the MS, and send the revised MS back with an explanation of alterations, or why you disagree with suggestions. It will not go out for another review. I'll look at your revisions and make a decision, so the turnaround on my end will be rapid. It is my view that you will be able to successfully deal with these suggestions.

Reviewer 1 ·

Basic reporting

acceptable

Experimental design

acceptable

Validity of the findings

accetpable

Additional comments

The authors have corrected the errors with reporting and statistics and have revised the text so that is is largely suitable. This discussion is more nuanced, but is still not adequate. Here is the fundamental problem: at a sufficient inter individual spacing, the plume filaments will not interact, meaning the role of mixing will be to dilute the individual filaments. A time averager would not be strongly affected by this because the time average concentration at this smaller scale would not change drastically whereas a creature relying on encoding filaments might be strongly and negatively affected. It might even be that the greater mixing would result in a time averager being exposed to the signal and increase success, which would be consistent with the observations of Wilson and Weissburg. (Fish, incidentally, probably are not time averaging- studies have shown they "sniff"-e.g. take discrete samples (e.g. Nevitt). Thus. the exact role of group size depends in the interplay between inter individual spacing, flow and the sensory strategy of the forager. I think this is what needs to be articulated. The results of this study are most applicable, in this reviewer's opinion, to situations where the prey are densely packed so that individual odor filaments are less important, even if the forager relies on time averaging. Still this is important-for instance bivalve aggregations, etc.I believe the discussion on these issues (lines 185-189, 206) needs to mention these complications.

A few minor suggestions noted by comments in the pdf.

Annotated reviews are not available for download in order to protect the identity of reviewers who chose to remain anonymous.

---

## Round 0.3 · accepted · Accept

· Academic Editor

Accept

Your revisions adequately addressed the remaining reviewer concerns. Thanks for working through this to create an improved publication.